# OpenReview forum: "Annotation-Efficient Honesty Alignment via Confidence Elicitation and Calibration"
_ICLR.cc/2026/Conference — ICLR 2026 Poster_

### Official Review · Reviewer_R9C2 · 2025-10-29

**Soundness:** 3
**Presentation:** 3
**Contribution:** 3
**Rating:** 6
**Confidence:** 3

**Summary:**

This paper proposed an annotation-efficient framework EliCal for the honesty alignment of LLMs.
And this paper also establish a large-scale benchmark HonestyBench for evaluating the performance of various confidence calibration methods. Notably, the proposed EliCal adopts a pretrain-then-finetune paradigm, which significantly reduces the need for labeled data by enhancing the generalization ability of confidence calibration.

**Strengths:**

1) This paper is well-written and easy to read.
2) After Elicitation-Then-Calibration, the model could express its confidence before generating any response token, which can efficiently reduce the computational overhead of sampling and consistency checking.
3) The method proposed by the authors is very simple and effective, while also easy to scale.

**Weaknesses:**

1) When constructing the Consistency Data, EliCal uses the greedy-search answer as the most confident response (Equation 5). However, the generation probability of the greedy-search answer is not necessarily the globally maximal. Using beam search to $\tilde r$ might be more reasonable.

2) It seems that authors does not explain how to evaluate whether two responses are semantically consistent in line 168 - line 175.

**Questions:**

1) In Equation 7, for $r$ from a sampled set $\hat{\mathcal
R} $, it is easy to obtain $p(r | q)$. Does using the estimation method in Equation 7 introduce additional error?

2) In Figure 2, the self-consistency confidence is highly correlated with true capabilities (Spearman coefficient = 0.789). Perhaps isotonic regression or platt scaling would be sufficient for calibration. It is recommended that the authors include experimental results using these simple baselines.

3) In line 252, does ETC means ELICITATION-THEN-CALIBRATION (ELICAL)?

4) The authors should show the performance of each model before and after confidence calibration in Table 3.

---

> ### Author Response · Authors · 2025-11-24
> **Author Response (1/2)**
>
> Thank you very much for reviewing our paper and for providing so many valuable suggestions. We also appreciate your recognition of our writing and our proposed method, including its simplicity, effectiveness, and ease of scaling. In response to your concerns, we provide the following replies.
>
> ---
>
> > **Response to Weakness1**.
>
> We agree that the greedy-search answer does not necessarily correspond to the global maximum of the generation probability, and that beam search could also be a reasonable alternative. However, identifying the true global maximum probability is intractable in practice, and prior work [3,4,5] consistently adopts the greedy-search answer as the model’s most confident prediction for this reason.
> **Following this established practice**, EliCal uses the greedy-search answer when constructing the Consistency Data.
>
> ---
>
> > **Response to Weakness2**.
>
> As stated in lines 331–334 of the paper, **we use Qwen2.5-32B-Instruct to evaluate whether the two sentences are semantically consistent**. The specific prompt can be found in Figure 12 in the appendix. Using an LLM to judge semantic consistency is currently the most widely adopted and accurate method for this purpose.[1,2,4]
>
> ---
>
> > **Response to Question1**.
>
> Since the model’s generation probability for an answer cannot be computed directly, we rely on more samples to obtain a more accurate estimate. We conduct an ablation study on the number of samples K. As K increases, the value of Consis-Sem (Equation 7) also increases, but with diminishing returns. **We set K=20 in the paper to provide a reasonable balance between estimation accuracy and computational cost**.
>
> | K   | Methods| In-Domain Evaluation | | |||| OOD Evaluation   ||||||
> |-|--|---|-|-|--|--|--|--|--|--|--|--|--|
> ||| NQ   | TQ   | HQ   | 2Wiki | Pararel | Avg  | Squad | WQ   | CWQ  | MSQ  | PopQA | Avg  |
> | 2   | Consis-Sem | 75.42| 83.04| 73.13| 51.13 | 60.93  | 68.04| 62.39 | 70.52| 70.94| 63.55| 66.26 | 65.58|
> | 5   | Consis-Sem | 78.72| 87.76| 77.85| 53.14 | 62.05  | 71.44| 64.59 | 73.88| 74.99| 67.63| 68.69 | 68.29|
> | 10  | Consis-Sem | 79.70 | 89.4 | 79.41| 54.41 | 62.47  | 72.78| 65.68 | 75.25| 76.58| 69.7 | 69.82 | 69.54|
> | 20  | Consis-Sem | 80.68| 90.20| 80.12| 55.40 | 62.93  | 73.62| 66.16 | 76.26| 77.50| 70.76| 70.44 | 70.20|
>
> ---
>
> > **Response to Question2**.
>
> Thank you for the suggestion. We have evaluated whether standard calibration baselines can effectively calibrate the self-consistency confidence.
>
> **[Isotonic Regression and Platt Scaling].**
>
> We additionally applied **Isotonic Regression** and **Platt Scaling** to the raw self-consistency confidence. The AUROC **remains essentially unchanged relative to the uncalibrated scores**. This is expected because AUROC evaluates ranking quality:
> 1. Isotonic Regression enforces a monotone non-decreasing mapping and therefore cannot change the ranking.
> 2. Platt Scaling fits a logistic function that is typically monotone increasing; in our experiments, the learned function also preserves the original ordering.
>
> Therefore, although these baselines can adjust numerical probabilities, **they cannot improve the discrimination between correct and incorrect predictions, and do not achieve effective calibration for this task**.
>
>
> > 1k and 560k indicate the number of training examples.
>
> | Methods| In-Domain Evaluation | | | |  | | OOD Evaluation   | | | |  | |
> |----|--|--|--|--|---|--|--|--|--|--|---|--|
> || NQ | TQ| HQ| 2Wiki | Pararel| AVG   | Squad | WQ| CWQ   | MSQ   | PopQA  | AVG   |
> | Consis-Sem | 80.68| 90.20 | 80.12 | 55.40 | 62.93  | 73.62 | 66.16 | 76.26 | 77.50 | 70.76 | 70.44  | 70.20 |
> | Isotonic Regression(1k)| 80.48| 89.90 | 79.87 | 55.39 | 62.82  | 73.45 | 66.00 | 76.03 | 77.16 | 70.95 | 70.34  | 70.07 |
> | Platt Scaling (1k)| 80.68| 90.20 | 80.12 | 55.40 | 62.93  | 73.62 | 66.16 | 76.26 | 77.50 | 70.76 | 70.44  | 70.20 |
> | Isotonic Regression(560k)| 80.68  | 90.20 | 80.12 | 55.40 | 62.93  | 73.62 | 66.16 | 76.26 | 77.50 | 70.76 | 70.44  | 70.20 |
> | Platt Scaling (560k)   | 80.68| 90.20 | 80.12 | 55.40 | 62.93  | 73.62 | 66.16 | 76.26 | 77.50 | 70.76 | 70.44  | 70.20 |
>
> **[Linear mapping ablation].**
>
> As reported in lines 469–476 and Figure 10, replacing LoRA + classification head with a single linear layer substantially degrades performance. With only 1k labeled samples, the AUROC is reduced by approximately 10 points compared to our full method (Figure 10 vs. Figure 5). This indicates that **a simple mapping is insufficient for effective calibration.**

---

> > ### Author Response · Authors · 2025-11-24
> > **Author Response (2/2)**
> >
> > > **Response to Question3**.
> >
> > Yes, sorry for the confusion. It should be EliCal, and we have already corrected this in the paper.
> >
> >
> > > **Response to Question4**.
> >
> > Thank you for the suggestion. Table 4 in the appendix already reports the performance of each model before and after confidence calibration. To improve clarity, we will merge Tables 3 and 4 in the revised version so that all results are presented together.
> >
> > ---
> >
> > We sincerely thank you again for reviewing our paper, for your recognition, and for providing many valuable suggestions. We also appreciate your efforts in helping us improve the completeness of our work.
> >
> > ---
> >
> > **References.**
> >
> > [1] Semantic Uncertainty: Linguistic Invariances for Uncertainty Estimation in Natural Language Generation. ICLR 2023
> >
> > [2] SAC3: Reliable Hallucination Detection in Black-Box Language Models via Semantic-aware Cross-check Consistency. EMNLP 2023
> >
> > [3] R‑Tuning: Instructing Large Language Models to Say ‘I Don’t Know’. NAACL 2024
> >
> > [4] Alignment for Honesty. NeurIPS 2024
> >
> > [5] Towards Fully Exploiting LLM Internal States to Enhance Knowledge Boundary Perception. ACL 2025

---

> > > ### Author Response · Authors · 2025-11-28
> > > **Official Comment by Authors**
> > >
> > > Dear Reviewer R9C2,
> > >
> > > As we approach the end of the discussion period, we kindly invite you to share any additional thoughts regarding our response to your concerns above. We sincerely appreciate your efforts and valuable feedback thus far.
> > >
> > > Best regards,
> > >
> > > The Authors

---

### Official Review · Reviewer_wtAu · 2025-11-02

**Soundness:** 3
**Presentation:** 3
**Contribution:** 3
**Rating:** 6
**Confidence:** 2

**Summary:**

The paper addresses the critical issue of honesty alignment in large language models, which is the ability of these models to accurately recognize their knowledge boundaries and express calibrated confidence. The authors introduce a two-stage framework called Elicitation-Then-Calibration (EliCal). This innovative approach first elicits the model's internal confidence using inexpensive self-consistency supervision and subsequently calibrates this confidence with a small set of correctness annotations. This method significantly reduces the annotation requirements while improving the model's generalization across various tasks.

**Strengths:**

1. The paper presents a novel approach to reward modeling by framing it as a text generation task. This innovative perspective allows for the generation of structured justifications for actions, which significantly enhances the interpretability of agent decisions.

2. The authors provide a thorough comparison of WebArbiter against existing models, showcasing its superior performance across various benchmarks.

3. This paper addresses critical limitations in existing reward models and provides a more reliable framework for web agents.

**Weaknesses:**

1. The experiments primarily evaluate relatively small and weak open-source models (mainly LLaMA). It remains unclear whether the conclusions generalize to larger models (e.g., those with more than 30B parameters).

2. In Appendix G, there appears to be a missing or incomplete figure (indicated by “??”).

3. In the experiments, the authors use an LLM-as-a-judge approach to evaluate QA performance but do not provide any justification or evidence for its reliability.

**Questions:**

see above

---

> ### Author Response · Authors · 2025-11-24
> **Author Response**
>
> Thank you very much for reviewing our paper and for acknowledging the value of our work in the summary. We respectfully inquire whether the Strengths section was meant for a different paper, as it seems unrelated to our work.
>
> Regarding the concerns raised in the Weaknesses section, we provide the following responses.
>
> ---
>
> > **Response to Weakness1**.
>
> We have added experiments with **Qwen2.5-32B-Instruct** on HonestyBench, and the results are shown in the table below. We found that **EliCal(1k) still significantly outperforms Cal-Only(1k), indicating that our method remains effective on larger models**.
>
> > The maximum values among the first four methods are bolded.
>
> | Methods   | In-Domain Evaluation |||| || OOD Evaluation  |||| ||
> |-|----|-|-|-|--|-|-|-|-|-|--|-|
> |    | NQ     | TQ    | HQ    | 2Wiki | Pararel| AVG   | Squad | WQ    | CWQ   | MSQ   | PopQA  | AVG   |
> | Consis-Sem| 80.18  | 89.07 | 80.78 | 56.62 | 73.11  | 74.57 | 66.89 | 74.97 | 76.24 | 71.31 | 74.66  | 72.10 |
> | Eli-Only  | 78.26  | 87.31 | 79.14 | 52.55 | 73.98  | 72.26 | 64.64 | 75.86 | 75.34 | 63.67 | 79.55  | 72.90 |
> | Cal-Only (1k)    | 75.15  | 81.12 | 79.63 | 78.37 | 80.67  | 79.31 | 73.64 | 72.99 | 76.69 | 69.27 | 87.47  | 79.62 |
> | EliCal (1k)      | **81.53**  | **88.14** | **84.73** | **81.74** | **83.98**  | **84.39** | **78.88** | **78.08** | **80.41** | **77.10** | **90.71**  | **84.01** |
> | Cal-Only (560k)  | 85.19  | 89.59 | 86.18 | 85.17 | 88.52  | 86.95 | 82.38 | 79.22 | 81.78 | 81.91 | 91.32  | 85.97 |
> | EliCal (560k)    | 85.11  | 89.91 | 86.26 | 85.22 | 89.05  | 87.12 | 82.86 | 80.38 | 82.41 | 83.04 | 91.55  | 86.45 |
>
>
>
> ---
>
> > **Response to Weakness2**.
>
> Thank you very much for your careful review. All references to figures have been carefully checked and updated.
>
> ---
>
> > **Response to Weakness3**.
>
> **[Why LLMs Are Used to Determine Answer Correctness]**.
>
> 1. Traditional metrics like F1 measure term-level overlap between model-generated and ground-truth answers, but they do not reliably capture semantic similarity, as they are sensitive to word forms and irrelevant terms.
> 2. **LLMs have been shown to effectively measure semantic consistency between sentences [1][2][3] and are widely used for answer correctness evaluation**. We follow this approach, using Qwen2.5-32B-Instruct to ensure effectiveness within reasonable computational cost.
>
> **[Validation via Human Evaluation]**.
>
> To validate the reliability of the judgments, we conducted human evaluation. We asked a student currently pursuing a master’s degree to evaluate the first 50 NQ-test samples. **Disagreements occurred in 4 cases, yielding 92\% consistency between human and model judgments, demonstrating the reliability of our approach**.
>
> ---
>
> We would like to thank you again for reviewing our paper and for providing many valuable suggestions. These recommendations have helped improve the completeness of the paper, and we hope our responses are helpful.
>
> ---
>
> **References**.
>
> [1] Semantic Uncertainty: Linguistic Invariances for Uncertainty Estimation in Natural Language Generation. ICLR 2023
>
> [2] SelfCheckGPT: Zero-Resource Black-Box Hallucination Detection for Generative Large Language Models. EMNLP 2023
>
> [3] LLM-as-a-Judge: Reassessing the Performance of LLMs in Extractive QA. Arxiv 2025

---

> > ### Author Response · Authors · 2025-11-28
> > **Official Comment by Authors**
> >
> > Dear Reviewer wtAu,
> >
> > As we approach the end of the discussion period, we kindly invite you to share any additional thoughts regarding our response to your concerns above. We sincerely appreciate your efforts and valuable feedback thus far.
> >
> > Best regards,
> >
> > The Authors

---

### Official Review · Reviewer_6wW6 · 2025-11-03

**Soundness:** 2
**Presentation:** 3
**Contribution:** 2
**Rating:** 2
**Confidence:** 4

**Summary:**

To avoid exhausting human annotation in honest alignment, this work proposes an Elicitation-Then-Calibration strategy, which measures the self-consistency of LLMs’ multiple generations as an alternative signal for output correctness estimation and evaluation. The authors adopt a linear layer with the last hidden states for output consistency prediction. Experiments on ten datasets against seven baselines verify the effectiveness of the proposed method.

**Strengths:**

1. The authors regard the self-consistency of LLM outputs as an indicator of its confidence, which is further leveraged to assess the correctness of its generations. According to this setting, large-scale human annotation and LLM fine-tuning can be avoided.
2. Extensive comparison of 7 representative methods, including both training-free and traning-based, verifies the effectiveness on the HonestBench.

**Weaknesses:**

1. The method fundamental premise lacks sufficient theoretical justification and empirical validation regarding its effectiveness and robustness.
2. Important design factors such as the sampling number k, the necessity and contribution of Stage 2 (Confidence Calibration), and the accuracy of the linear-layer-based self-consistency prediction (e.g., MSE) are not adequately discussed and experimentally analyzed.
3. The evaluation lacks comparisons with recent baselines and omits results using true self-consistency (Eq. 7) versus predicted values (Eq. 8).

**Questions:**

1. My primary concern is regarding the actual effectiveness and validity of the basic assumption: the confidence of generation can be quantified through the self-consistency among the multiple outputs, given that it is the most important premise regarding the technical soundness of your method. Although the authors provide some reference for self-defense, more theoretical analysis and experiments are necessary to support this conclusion.
2. How accurate or reliable is the self-consistency in estimating the correctness of LLM generations? Figure 2 does not present this relationship clearly or directly. Moreover, it is uncertain whether this prediction accuracy can lead to tangible performance gains in practical scenarios. As far as I am concerned, in most cases, unexpected or alignment-required outputs constitute only a small proportion, and thus, the potential improvement might be offset by incorrect predictions of the self-consistency.
3. Also, the number of sampled generations (i.e., k) seems important in your method. However, no discussion and experiments regarding this hyperparameter are provided.
4. Stage 2 Confidence-Calibration requires detailed elaboration. I am not sure of the necessity and improvements of this stage to the final performance.
5. How about the prediction accuracy of self-consistency through the linear layer? I recommend that the authors report the MSE metrics in experiments.
6. How would the performance on HonestBench change if the true self-consistency, computed directly from Equation (7), were used instead of the predicted values obtained from Equation (8)?
7. I suggest the authors supplement more recent baselines for a fair and comprehensive performance comparison.
8. I recommend the authors release the relevant data and code to enhance the reproducibility of the method.
9. Citation error: “Figure ??” (Appendix G, p. 20). Please perform a thorough proofreading pass and fix all broken cross-references.

---

> ### Author Response · Authors · 2025-11-24
> **Author Response (1/3)**
>
> Thank you for reviewing our paper and for offering so many valuable suggestions. In response to your concerns, we provide the following replies.
>
> ---
>
> > **Response to “the necessity and contribution of Stage 2 (Confidence Calibration)” and “omitting results using true self-consistency (Eq. 7) versus predicted values (Eq. 8)”**.
>
> Apologies for any confusion caused. We think there may have been some misunderstanding regarding our experimental setting and result analysis, as these points are discussed in the paper. We hope the following clarifications could help.
>
> For clarity, we extract a subset of the results from Table 2 of the paper, as shown below. As described in Lines 349–357 of the paper, these methods correspond to the following settings:
> - **Consis-Sem**: The true consistency computed from multiple samples using Eq. 7
> - **Eli-Only**: The model after Stage 1 Confidence Elicitation (using Eq. 8), where Consis-Sem serves as the supervision signal for this stage
> - **EliCal**(1k): Building on Eli-Only by using 1k labeled examples to perform Stage 2 Confidence Calibration
>
> |Methods|In-Domain Evaluation||||||OOD Evaluation||||||
> |-|-|-|-|-|-|-|-|-|-|-|-|-|
> ||NQ|TQ|HQ|2Wiki|Pararel|Avg |Squad|WQ|CWQ|MSQ|PopQA|Avg|
> |Consis-Sem|80.68|90.20|80.12|55.40|62.93 |73.62|66.16|76.26|77.50|70.76|70.44|70.20|
> |Eli-Only |77.86|86.23|77.27|54.36|62.05 |71.19|60.66|76.61|74.77|66.56|74.60|69.66|
> |EliCal (1k) |**82.38**|**87.51**|**84.48**|**82.05**|**84.31** |**84.36**|**78.48**|**80.11**|**79.85**|**78.09**|**91.74**|**84.47**|
>
> **[The necessity and contribution of Stage 2]**.
>
> Comparing EliCal(1k) with Eli-Only, we observe that EliCal(1k) outperforms by about 13\% on in-domain datasets and about 15\% on OOD datasets, **demonstrating the necessity of the second stage**.
>
> **[True self-consistency (Eq. 7) versus predicted values (Eq. 8)]**.
>
> As mentioned in Lines 432–435 of the paper, comparing Eli-Only and Consis-Sem shows that the predicted values perform similarly to the true self-consistency. This provides a solid foundation for the second stage.
>
> ---
>
> > **Response to Weakness 1 & Question 1**.
>
> We agree that the effectiveness of self-consistency–based confidence is fundamental to our method. We would like to say that this is not an assumption introduced in our work, but rather a concept whose utility has been widely demonstrated in honesty alignment research [1–4].
> Providing formal theoretical guarantees is challenging due to model complexity and the inaccessibility of training data; thus, our focus is on a clear **conceptual motivation supported by strong empirical evidence**.
>
> [**Conceptual Justification**].
>
> Prior studies [1–4] suggest that a model is more likely to produce correct answers when its probability distribution over possible outputs is concentrated, and more likely to err when the distribution is dispersed. This principle implies that estimating a model’s generation probability for an answer can serve as a confidence signal.
> Three common approaches to measure this probability are:
> 1. Token-Probability-Based Confidence [1]: Multiplying token-level probabilities of the entire response. Sensitive to surface forms, so semantically equivalent answers may differ in score.
> 2. Verbalized Confidence [5]: Asking the model to report confidence in words. Often inaccurate and overconfident.
> 3. Self-Consistency-Based Confidence [1–4]: Sampling multiple outputs and measuring semantic agreement. This method directly captures certainty over the semantic content of the answer and **has been shown to correlate most strongly with answer correctness**.
>
> Thus, self-consistency provides a natural and reliable measure of model confidence (Lines 38–43).
>
> [**Empirical Validation**].
>
> 1. We validate the correlation between confidence based on semantic self-consistency (Consis-Sem) and model capability in the paper (Table 4 and Figure 4). For convenience, we present Figure 4 in tabular form, showing the average AUROC across three models on the in-domain evaluation datasets.
> It can be seen that **Consis-Sem performs the best**, indicating that the confidence of generation can be quantified through the self-consistency among multiple outputs.
> |Methods|AUROC|
> |-|-|
> |Prob|65.6|
> |N-Prob|69.5|
> |Verbal-0|68.8|
> |Verbal-10|70.2|
> |Consis-Lex|70.7|
> |Consis-Sem|**73.0**|
>
> 2. In addition, Table 2 shows the performance of Qwen2.5-7B-Instruct trained with the first stage (EliCal) versus without it (Cal-Only). With the same amount of labeled data (1k), EliCal significantly outperforms Cal-Only, **demonstrating the effectiveness of Consis-Sem as the training signal in the first stage**.
>
> |Methods|In-Domain Evaluation||||||OOD Evaluation||||||
> |-|-|-|-|-|-|-|-|-|-|-|-|-|
> ||NQ|TQ|HQ|2Wiki|Pararel|Avg|Squad|WQ|CWQ|MSQ|PopQA|Avg|
> |Cal-Only(1k)|72.19|68.75|74.34|76.17|78.61|73.41|71.59|71.48|69.33|66.96|86.13|77.32|
> |EliCal(1k)|**82.38**|**87.51**|**84.48**|**82.05**|**84.31**|**84.36**|**78.48**|**80.11**|**79.85**|**78.09**|**91.74**|**84.47**|

---

> > ### Author Response · Authors · 2025-11-24
> > **Author Response (2/3)**
> >
> > > **Response to Weakness2**.
> >
> > **[Ablation on Sampling Number K]**.
> >
> > An ablation study was conducted on the number of samples K used to compute self-consistency–based confidence with Qwen2.5-7B-Instruct. In Stage 1, $K\in\{2,5,10,20\}$ was evaluated. The key observations are as follows:
> > 1. **Consis-Sem**: Estimation quality consistently improves with larger K.
> > 2. **Eli-Only**: Performance varies only slightly. Notably, even at K=2, despite the additional noise in Consis-Sem, the signal remains sufficiently informative, enabling the model to learn to express its internal confidence from large-scale unlabeled data.
> > 3. **EliCal**: Performance is highly stable across different K, **demonstrating robustness**.
> >
> > | K| Methods | In-Domain Evaluation |||||| OOD Evaluation ||||||
> > |-|--|----|-|-|-|-|-|----|-|-|-|-|-|
> > || | NQ| TQ| HQ| 2Wiki | Pararel | AVG| Squad | WQ| CWQ| MSQ| PopQA | AVG|
> > | 2| Consis-Sem| 75.42| 83.04| 73.13| 51.13 | 60.93| 68.04| 62.39 | 70.52| 70.94| 63.55| 66.26 | 65.58|
> > || Eli-Only| 78.46| 86.03| 77.30| 52.48 | 63.59| 70.70| 60.61 | 77.20| 75.24| 64.67| 75.32 | 69.90|
> > || EliCal (1k)| 82.37| 87.32| 84.60| 82.13 | **85.04**| 84.41| 79.03 | **81.04**| 79.72| 78.45| 91.94 | **84.80**|
> > | 5| Consis-Sem| 78.72| 87.76| 77.85| 53.14 | 62.05| 71.44| 64.59 | 73.88| 74.99| 67.63| 68.69 | 68.29|
> > || Eli-Only| 78.86| 86.09| 77.82| 54.10 | 65.45| 71.54| 62.34 | 77.42| 76.12| 67.13| 77.92 | 71.88|
> > || EliCal (1k)| 82.49| 87.46| 84.46| **82.37** | 84.98| **84.52**| **79.05** | 80.60| 79.17| **78.70**| 91.95 | 84.74|
> > | 10| Consis-Sem| 79.70| 89.40| 79.41| 54.41 | 62.47| 72.78| 65.68 | 75.25| 76.58| 69.70| 69.82 | 69.54|
> > || Eli-Only| 78.83| 86.26| 77.36| 54.17 | 63.41| 71.36| 61.51 | 77.61| 75.58| 66.52| 76.05 | 70.71|
> > || EliCal (1k)| **82.58**| 87.56| **84.64**| 82.22 | 84.59| 84.51| 78.88 | 80.87| 79.28| 78.05| **91.98** | 84.68|
> > | 20| Consis-Sem| 80.68|**90.20**| 80.12| 55.40 | 62.93| 73.62| 66.16 | 76.26| 77.50| 70.76| 70.44 | 70.20|
> > || Eli-Only| 77.86| 86.23| 77.27| 54.36 | 62.05| 71.19| 60.66 | 76.61| 74.77| 66.56| 74.60 | 69.66|
> > || EliCal (1k)| 82.38| 87.51| 84.48| 82.05 | 84.31| 84.36| 78.48 | 80.11| **79.85**| 78.09| 91.74 | 84.47|
> >
> > Larger K may further refine Consis-Sem estimation, but the gains plateau quickly. Using K=20 already provides an accurate approximation while keeping computation reasonable. Moreover, EliCal remains stable under the minor fluctuations of Consis-Sem, which is why experiments with K > 20 are omitted.
> >
> > Thank you for your suggestion. These findings indicate that **it requires only a small number of samples to construct effective self-consistency–based supervision**, further improving the efficiency. This ablation will be included in the revised version.
> >
> > **[Evaluation of Linear-Layer-Based Self-Consistency Prediction]**.
> >
> > The Linear-Layer-Based Self-Consistency Prediction was further evaluated on Qwen2.5-7B-Instruct by computing the MSE between the predicted values and the true self-consistency–based confidence. For comparison, the **MSE** obtained using the LoRA + Linear head configuration from the main paper was also reported.
> > - The results show that Eli-Only (Linear) yields nearly double the MSE of Eli-Only, indicating that **relying solely on a linear transformation leaves substantial room for improvement**.
> >
> > > MSE(Predicted Confidence, Consis-Sem)
> >
> > | Methods| In-Domain Evaluation |||||| OOD Evaluation ||||||
> > |--|-|-|-|-|-|-|-|-|-|-|-|-|
> > | | NQ| TQ| HQ| 2Wiki | Pararel | AVG| Squad | WQ| CWQ| MSQ| PopQA | AVG|
> > | Eli-Only (Linear)| 0.09 | 0.11 | 0.10 | 0.06| 0.10| 0.09 | 0.10| 0.10 | 0.11 | 0.10 | 0.11| 0.11|
> > | Eli-Only| **0.06** | **0.07** | **0.06** | **0.03**| **0.07**| **0.05** | **0.07**| **0.06** | **0.07** | **0.05** | **0.06**| **0.06**|
> >
> > AUROC values after Stage 1 and Stage 2 are also presented below. The results consistently show that training with only a linear head underperforms both during Elicitation and Calibration.
> > This **highlights the importance of using LoRA to enable deeper interaction with the model's internal states**.
> >
> > > AUROC(Predicted Confidence, Accuracy)
> >
> > | Methods| In-Domain Evaluation |||||| OOD Evaluation ||||||
> > |---|-|-|-|-|-|-|-|-|-|-|-|-|
> > | | NQ| TQ| HQ| 2Wiki | Pararel | AVG| Squad | WQ| CWQ| MSQ| PopQA | AVG|
> > | Consis-Sem| 80.68| 90.20| 80.12| 55.40 | 62.93| 73.62| 66.16 | 76.26| 77.50| 70.76| 70.44 | 70.20|
> > | Eli-Only (Linear)| 73.26| 76.32| 69.63| 45.38 | 62.06| 63.33| 57.87 | 72.40| 67.67| 60.06| 71.93 | 66.10|
> > | Eli-Only| 77.86| 86.23| 77.27| 54.36 | 62.05| 71.19| 60.66 | 76.61| 74.77| 66.56| 74.60 | 69.66|
> > | EliCal (1k, Linear)| 73.76| 76.25| 72.84| 59.04 | 69.81| 69.13| 63.86 | 72.91| 70.25| 62.13| 79.91 | 71.96|
> > | EliCal (1k)| 82.38| 87.51| 84.48| 82.05 | 84.31| 84.36| 78.48 | 80.11| 79.85| 78.09| 91.74 | 84.47|

---

> > > ### Author Response · Authors · 2025-11-24
> > > **Author Response (3/3)**
> > >
> > > > **Response to Weakness3 (Regarding More Recent Baselines)**.
> > >
> > > Since correctness-based supervision is widely regarded as an effective approach for honesty alignment[4,6]—and the proposed method is designed to optimize this fine-tuning process—**the primary comparison target is Calibration-Only (Cal-Only)**. To provide a more complete evaluation, two additional training-based post-hoc calibration methods were included.
> > > 1. **Thermometer**[6], which learns a mapping from the model’s internal states to a temperature value on the training set and then predicts a temperature at test time to adjust model confidence [6].
> > > 2. **DACA**[7], which calibrates the logits of a model after SFT and RLHF (Qwen2.5-7B-Instruct) using a pre-trained base model (Qwen2.5-7B) through a learned temperature [2].
> > > Both baselines were trained using 1,000 randomly sampled instances from each of the five datasets in HonestyBench-Train (5,000 examples in total).
> > >
> > > Since these two methods use ECE as the evaluation metric in their original papers, we report both AUROC and ECE. The results on Qwen2.5-7B-Instruct are shown below.
> > > Results show that **EliCal(1k) outperforms both methods in terms of both ECE and AUROC**. Although the two baselines are effective in decreasing ECE—i.e., making the overall confidence closer to the empirical accuracy—they lack sufficient discriminative power between correct and incorrect samples.
> > >
> > > |Methods|In-Domain Evaluation (AUROC)||||||OOD Evaluation (AUROC)||||||
> > > |-|-|-|-|-|-|-|-|-|-|-|-|-|
> > > ||NQ|TQ|HQ|2Wiki|Pararel|Avg|Squad|WQ|CWQ|MSQ|PopQA|Avg|
> > > |Thermometer[7]|58.15|63.38|58.08|55.24|67.58 |59.48|60.23|62.90|56.98|61.90|68.76|63.88|
> > > |DACA[8]|61.69|72.54|66.39|65.03|71.06 |67.70|66.79|65.71|62.62|73.98|76.40|70.98|
> > > |Cal-Only(1k)|72.19|68.75|74.34|76.17|78.61 |73.41|71.59|71.48|69.33|66.96|86.13|77.32|
> > > |EliCal(1k)|**82.38**|**87.51**|**84.48**|**82.05**|**84.31** |**84.36**|**78.48**|**80.11**|**79.85**|**78.09**|**91.74**|**84.47**|
> > >
> > > |Methods|In-Domain Evaluation(ECE)||||||OOD Evaluation(ECE)||||||
> > > |---|-|-|-|-|-|-|-|-|-|-|-|-|
> > > |NQ|TQ |HQ |2Wiki|Pararel|Avg |Squad|WQ |CWQ|MSQ|PopQA|Avg |
> > > |Thermometer[7]|0.06|**0.05**|0.04|0.05|0.04|0.06|**0.05**|**0.05**|**0.01**|**0.06**|0.10|0.07|
> > > |DACA[8] |0.09|0.05|0.10|0.14|0.10|0.10|0.09|0.08|0.06|0.06|0.15|0.11|
> > > |Cal-Only(1k)|0.12|0.16|0.18|0.13|0.13|0.15|0.17|0.15|0.16|0.12|0.07|0.12|
> > > |EliCal(1k) |**0.05**|0.06|**0.03**|**0.05**|0.08|**0.05**|0.10|0.06|0.05|0.08|**0.02**|**0.06**|
> > >
> > > ---
> > > > **Response to Question1: `See Response to Weakness1`**.
> > >
> > > ---
> > >
> > > > **Response to Question2**.
> > >
> > > 1. How accurate or reliable is the self-consistency in estimating the correctness of LLM generations -> See `Response to Weakness1`
> > > 2. We respectfully disagree with your claim that "in most cases, unexpected or alignment-required outputs constitute only a small proportion." The hallucination problem in current LLMs remains a significant challenge. If models could provide reliable confidence, it would enable us to determine when to trust their outputs. As shown in the appendix of the paper, in Table 3, the QA performance of the three test models did not exceed 50\%,
> > > and in many cases, the models provided incorrect answers. Therefore, we believe that honesty alignment is of utmost importance.
> > >
> > > ---
> > >
> > > > **Response to Question3 & Question4 & Question5: See `Response to Weakness2`**.
> > >
> > > ---
> > >
> > >
> > > > **Response to Question6 & Question7: See `Response to Weakness3`**
> > >
> > > ---
> > >
> > > > **Response to Question8**.
> > > We have organized the data, code, and a one-click run script. We will provide the code link after the conclusion of the double-blind review process.
> > >
> > > ---
> > >
> > > > **Response to Question9**: Thank you for your thorough review. We have checked and made the necessary revisions.
> > >
> > > ---
> > >
> > > We sincerely thank you again for reviewing our paper and providing so many valuable suggestions. We greatly appreciate your efforts in helping improve the completeness of our work.
> > >
> > > ---
> > >
> > > **References.**
> > >
> > > [1] Semantic Uncertainty: Linguistic Invariances for Uncertainty Estimation in Natural Language Generation. ICLR 2023
> > >
> > > [2] SelfCheckGPT: Zero-Resource Black-Box Hallucination Detection for Generative Large Language Models. EMNLP 2023
> > >
> > > [3] SAC3: Reliable Hallucination Detection in Black-Box Language Models via Semantic-aware Cross-check Consistency. EMNLP 2023
> > >
> > > [4] Alignment for Honesty. NeurIPS 2024
> > >
> > > [5] When Do LLMs Need Retrieval Augmentation? Mitigating LLMs' Overconfidence Helps Retrieval Augmentation. ACL 2024
> > >
> > > [6] R-Tuning: Instructing Large Language Models to Say `I Don't Know'. NAACL 2024.
> > >
> > > [7] Thermometer: Towards Universal Calibration for Large Language Models. ICML 2024.
> > >
> > > [8] Your Pre-trained LLM is Secretly an Unsupervised Confidence Calibrator. NeurIPS 2025.

---

> ### Author Response · Authors · 2025-11-28
> **Official Comment by Authors**
>
> Dear Reviewer 6wW6,
>
> As we approach the end of the discussion period, we kindly invite you to share any additional thoughts regarding our response to your concerns above. We sincerely appreciate your efforts and valuable feedback thus far. You may have unintentionally overlooked some of the experimental results earlier, and we have provided clarification in the rebuttal. Regarding the theoretical justification and empirical validation of the assumptions you mentioned, we have added further explanations. In addition, we have included the ablation studies you requested, as well as more recent baseline methods. We look forward to your response!
>
> Best regards,
>
> The Authors

---

> > ### Comment · Reviewer_6wW6 · 2025-11-28
> > **Most concerns are well addressed.**
> >
> > Thanks for the efforts made by the authors to address the raised concerns. The scores will be adjusted positively when the system allows. Lastly, hope these updates would be included in the final version.

---

> > > ### Author Response · Authors · 2025-11-28
> > > **Author Response**
> > >
> > > Thank you for your quick response. We are glad to hear that most of your concerns have been addressed. We sincerely appreciate your contributions in helping us improve the completeness of the paper. We will incorporate all these updates into the final version. Have a good day！

---

### Official Review · Reviewer_tumR · 2025-11-04

**Soundness:** 3
**Presentation:** 3
**Contribution:** 2
**Rating:** 4
**Confidence:** 4

**Summary:**

The work proposes **EliCal (Elicitation-Then-Calibration)**, a two-stage framework for improving **honesty alignment** in large language models (LLMs)—the ability to express calibrated confidence aligned with correctness. Stage 1 (“Elicitation”) trains models to express internal confidence using *self-consistency* signals without annotations; Stage 2 (“Calibration”) fine-tunes this confidence using a small number of correctness-labeled examples. The work also introduces **HonestyBench**, a large-scale benchmark consolidating ten factual QA datasets (≈560k training and 70k evaluation pairs) annotated with both correctness and consistency signals. Empirical results show that EliCal attains near-upper-bound alignment performance using 0.18% of the labels required for full calibration, and can be generalized to unseen MMLU tasks.

**Strengths:**

1. **Well-motivated methodological design.** The proposed two-stage framework (EliCal) is well-motivated by the empirical correlation between self-consistency and correctness (Fig. 2), providing a principled basis for separating elicitation and calibration.
2. **Introduction of HonestyBench for universal evaluation.**  This benchmark combines 10 QA datasets, comprising 560k training samples and 70k evaluation samples, each annotated for correctness and self-consistency. It covers a variety of QA types—including single-hop, multi-hop, and template-generated questions—enabling systematic assessment both in-domain and out-of-domain.
3. **Significant experimental performance compared to the baselines.** The paper evaluates across three LLMs (Qwen2.5-7B, 14B; Llama-8B), showing consistent performance patterns (Table 2–4; Fig. 5–8), demonstrates strong cross-domain generalization to MMLU, and includes ablation studies probing the effects of elicitation dataset size and the limitations of the linear calibration head (Fig. 7–10).

**Weaknesses:**

1. **EliCal relies too heavily on the model's self-consistency performance.** However, self-consistency does not perform very well on all datasets [1], and for such datasets, this method still requires a large amount of labeled data for calibration.
2. **Even with LoRA used for training, the method is still very costly.** There is already a lot of work [2,3] applying token probability to open-ended QA tasks, and efficient post-hoc methods can be used for calibration, even without additional labeled data. In comparison, the cost of EliCal is much higher. Moreover, since it involves 2-stage training, this method is difficult to deploy on large-scale models in practice.
3. **The comparison of baselines may be unfair.** The purpose of EliCal is to assess the model's confidence in answering the current question correctly before the model generates an answer. However, in the baseline, both Prob and Verb require the model to generate an answer before providing a confidence score. I think there is a gap in the settings of these methods.
4. **The ECE experiment results are incomplete.** Since ECE is a very important metric in confidence calibration, in addition to the ECEs of EliCal and Cal-Only, could you also report the ECEs of the other baselines?

[1] Manakul, Potsawee, et al. "SelfCheckGPT: Zero-Resource Black-Box Hallucination Detection for Generative Large Language Models." *The 2023 Conference on Empirical Methods in Natural Language Processing*.

[2] Shen, Maohao, et al. "Thermometer: Towards Universal Calibration for Large Language Models." *Forty-first International Conference on Machine Learning*.

[3] Luo, Beier, et al. "Your Pre-trained LLM is Secretly an Unsupervised Confidence Calibrator." *The Thirty-Ninth Annual Conference on Neural Information Processing Systems.*

**Questions:**

1. **Can you compare ECE of the calibration method for token probability that I mentioned in the Weakness part with EliCal?** I'm curious how much the training method can improve compared to the post-hoc method.
2. **The setting of this article is a bit strange to me.** It is somewhat like uncertainty estimation and also like confidence calibration. In uncertainty estimation, we often use AUROC, while in confidence calibration, ECE is used as a metric. It is rare to evaluate both metrics at the same time. Could you explain to me the differences and connections between your task and these two tasks?

---

> ### Author Response · Authors · 2025-11-24
> **Author Response (1/3)**
>
> Thank you for your careful review of our paper and for many valuable suggestions. In response to your concerns, we provide the following replies.
> > **Response to Question2**.
>
> Apologies for any confusion caused. We first respond to Question 2 by clarifying the objective of our task and explaining why the paper primarily relies on AUROC for analysis.
>
> 1. **We focus on confidence calibration** — the goal is for the model to output confidence values that faithfully reflect its true ability, enabling **clear separation between questions it can answer correctly and those it cannot**.
> In our view, whether for uncertainty estimation or confidence calibration, both ECE and AUROC are applicable, as they focus on different aspects.
>
> 2. As you noted, ECE is a standard metric for confidence calibration. It aims to measure how close the model’s confidence is to its actual accuracy in a numerical sense, but this metric has a clear limitation: **ECE only captures average calibration quality**.
> For instance, consider two questions, A and B. The model outputs a confidence of 0.5 for both. It answers A incorrectly and B correctly. The average confidence is 0.5 and the average accuracy is also 0.5, resulting in an ECE of 0. Yet neither confidence value is meaningful — they fail to distinguish between correct and incorrect outcomes.
>
> 3. In contrast, **AUROC directly evaluates whether higher confidence is assigned to samples the model answers correctly**. It measures the discriminative power of the calibrated confidence. Therefore, we primarily rely on AUROC in our analysis. At the same time, to evaluate how well the overall confidence values match the empirical accuracy, we also report ECE results in the appendix.
>
> We appreciate your suggestion and will include a more detailed explanation of the evaluation metrics in the paper.
>
> ---
>
> > **Response to Weakness1**.
>
> We acknowledge that self-consistency (Consis-Sem in our paper) may perform suboptimally on certain datasets. Nevertheless, it remains relatively reliable among training-free signals and can still support calibration even when performance is suboptimal.
>
> 1. **Consis-Sem as a reliable training-free signal**.
> **Previous research [1-3], including the paper you cited [2], has shown** that self-consistency-based confidence generally outperforms other unsupervised signals such as random confidence, token-level probability/entropy, and proxy models. In that study[2], the authors measured sentence-level consistency using five different methods and found that the approach most similar to Consis-Sem—leveraging the LLM to judge consistency—achieved the best performance. **Our experiments (Table 2, Figure 4) further confirm** that Consis-Sem provides a stable and reliable signal (achieving high AUROC scores) among available training-free methods.
>
> 2. **Effective calibration even when suboptimal**.
> We extract part of the results from Table 2 and highlight the improvement of EliCal (1k) over Cal-Only (1k). **Even on datasets where Consis-Sem performs relatively poorly (e.g., 2Wiki, Pararel, Squad), EliCal (1k) still achieves a substantial improvement over Cal-Only (1k)**, although the gain is smaller than on other datasets. This demonstrates that even if the absolute correlation is not high, the relative ordering of instance-level confidence provided by Consis-Sem is sufficient to guide calibration, thereby reducing the amount of labeled data required.
> |Methods|NQ|TQ|HQ|2Wiki |Pararel|Squad |WQ|CWQ|MSQ|PopQA |
> |--|--|--|--|--|---|--|--|--|--|--|
> |**Consis-Sem**|80.68 |90.20 |80.12 |55.40 |62.93|66.16 |76.26 |77.50 |70.76 |70.44 |
> |**Cal-Only(1k)**|72.19 |68.75 |74.34 |76.17 |78.61|71.59 |71.48 |69.33 |66.96 |86.13 |
> |**EliCal(1k)** |82.38 |87.51 |84.48 |82.05 |84.31|78.48 |80.11 |79.85 |78.09 |91.74 |
> |**EliCal(1k) - Cal-Only(1k)** |10.19 |18.76 |10.14 |**5.88**|**5.70**|**6.89**|8.63|10.52 |11.13 |5.61|

---

> > ### Author Response · Authors · 2025-11-24
> > **Author Response (2/3)**
> >
> > > **Response to Weakness2**.
> >
> > [**Comparison with Recent Post-Hoc Methods**].
> >
> > Thank you for pointing out these two recent methods. We reproduced both on HonestyBench using Qwen2.5-7B-Instruct. The experimental settings and results are summarized below.
> >
> > 1. Settings:
> >  - **Thermometer**[4] performs temperature scaling on the generation probability of the answer and adaptively assigns a temperature for each task. This is done by learning a mapping from the model’s internal states to the temperature using the training set.
> >  - **DACA**[5] uses a pre-trained model (i.e., Qwen2.5-7B) to calibrate the logits of a model after SFT and RLHF (i.e., Qwen2.5-7B-Instruct) via a learned temperature.
> >  - For both methods, we followed the approach they outlined in their paper for handling free-form QA. We append the generated answer (the same as greedy-search answer in our paper) to the prompt, asking the model to judge whether the answer is correct by choosing between two options: A. Yes and B. No. For training, we sample 1,000 examples from each dataset in HonestyBench-Train (5,000 in total). All other settings follow the original configuration.
> >
> > 2. Results:
> >  - We present the experimental results in the two tables below. Considering your comment in Weakness 4, the first table shows the **ECE** scores for all methods, while the second table presents the **AUROC**.
> >  - **Although EliCal incurs a higher computational cost, it achieves significantly better performance**. Compared with EliCal (1k), **these post-hoc methods exhibit lower performance in terms of ECE and AUROC, particularly AUROC**. These methods tend to assign relatively high temperatures (mostly between 5 and 10), which uniformly reduces confidence for all questions and leads to lower ECE (as the overall confidence becomes closer to the empirical accuracy). However, this does not effectively enhance the ability of the confidence to discriminate between correct and incorrect predictions.
> >
> > |Methods|In-Domain Evaluation(ECE)||||||OOD Evaluation(ECE)||||||
> > |---|-|-|-|-|-|-|-|-|-|-|-|-|
> > |NQ|TQ |HQ |2Wiki|Pararel|Avg |Squad|WQ |CWQ|MSQ|PopQA|Avg |
> > |PC|0.39|0.53|0.32|0.31|0.46|0.40 |0.31|0.54|0.33|0.13|0.16|0.25|
> > |NPC|0.39|0.24|0.47|0.50 |0.32|0.39|0.46|0.24|0.46|0.65|0.60 |0.52|
> > |Verbal-0|0.41|0.28|0.44|0.41|0.28|0.37|0.42|0.29|0.44|0.46|0.57|0.48|
> > |Verbal-10|0.27|0.23|0.26|0.15|0.30 |0.22|0.35|0.32|0.36|0.46|0.36|0.36|
> > |Consis-Lex|0.07|0.07|0.16|0.17|**0.03**|0.12|0.14|0.07|0.13|0.32|0.26|0.20 |
> > |Consis-Sem|0.13|0.04|0.11|0.28|0.25|0.16|0.31|0.15|0.16|0.21|0.29|0.27|
> > |Thermometer[4]|0.06|0.05|0.04|0.05|0.04|0.06|**0.05**|**0.05**|**0.01**|**0.06**|0.10|0.07|
> > |DACA[5] |0.09|0.05|0.10|0.14|0.10|0.10|0.09|0.08|0.06|0.06|0.15|0.11|
> > |Elicitation|0.09|**0.03**|0.06|0.24|0.21|0.13|0.31|0.09|0.11|0.19|0.24|0.24|
> > |Cal-Only(1k)|0.12|0.16|0.18|0.13|0.13|0.15|0.17|0.15|0.16|0.12|0.07|0.12|
> > |EliCal(1k) |**0.05**|0.06|**0.03**|**0.05**|0.08|**0.05**|0.10|0.06|0.05|0.08|**0.02**|**0.06**|
> >
> > |Methods|In-Domain Evaluation (AUROC)||||||OOD Evaluation (AUROC)||||||
> > |-|-|-|-|-|-|-|-|-|-|-|-|-|
> > ||NQ|TQ|HQ|2Wiki|Pararel|Avg|Squad|WQ|CWQ|MSQ|PopQA|Avg|
> > |Consis-Sem|80.68|**90.20**|80.12|55.40|62.93 |73.62|66.16|76.26|77.50|70.76|70.44|70.20|
> > |Thermometer[4]|58.15|63.38|58.08|55.24|67.58 |59.48|60.23|62.90|56.98|61.90|68.76|63.88|
> > |DACA[5]|61.69|72.54|66.39|65.03|71.06 |67.70|66.79|65.71|62.62|73.98|76.40|70.98|
> > |Cal-Only(1k)|72.19|68.75|74.34|76.17|78.61 |73.41|71.59|71.48|69.33|66.96|86.13|77.32|
> > |EliCal(1k)|**82.38**|87.51|**84.48**|**82.05**|**84.31** |**84.36**|**78.48**|**80.11**|**79.85**|**78.09**|**91.74**|**84.47**|
> >
> > [**Training Cost Analysis**].
> >
> > We consider LoRA to be effective while incurring low training overhead.
> >
> > - Considering computational cost, as noted in your summary, we conducted an ablation study on LoRA in our paper, using only a linear head to predict confidence. As discussed in lines 469–477, the two-stage training remains effective for the linear head (compared to calibration-only), but its calibration performance is inferior to that achieved with LoRA. This indicates that **LoRA ensures more effective calibration**.
> > - Although the process involves two stages, the second stage requires only a small amount of data. **The overall training overhead is low and easy to scale**. For instance, for Qwen2.5-7B-Instruct, the trainable parameters account for only **0.28\% of** the model size, and the two-stage training takes just **two hours** on four B200 GPUs.

---

> > > ### Author Response · Authors · 2025-11-24
> > > **Author Response (3/3)**
> > >
> > > > **Response to Weakness3**.
> > >
> > > We are glad you noticed this. In fact, being able to **predict confidence before generating an answer is an advantage of our method**. Post-hoc methods can only estimate confidence after the answer is generated; thus, this comparison is not unfair to the baselines, but rather highlights that our method is **both effective and efficient**.
> > >
> > > ---
> > >
> > > > **Response to Weakness4**.
> > >
> > > We have supplemented the ECE results for each method, as shown in the table of `Response to Weakness2`. We will include all of these results in the paper.
> > >
> > > ---
> > >
> > > > **Response to Question1**: Can be found in `Response to Weakness2`'
> > >
> > > ---
> > >
> > > We sincerely thank you for the time and valuable suggestions you have provided to help improve the paper. Hope the responses are helpful.
> > >
> > > ---
> > >
> > > References:
> > >
> > > [1] Lorenz Kuhn et.al. Semantic Uncertainty: Linguistic Invariances for Uncertainty Estimation in Natural Language Generation. ICLR 2023
> > >
> > > [2] Potsawee Manakul et.al. SelfCheckGPT: Zero-Resource Black-Box Hallucination Detection for Generative Large Language Models. EMNLP 2023
> > >
> > > [3] Yuqing Yang et.al. Alignment for Honesty. NeurIPS 2024
> > >
> > > [4] Maohao Shen et al. Thermometer: Towards Universal Calibration for Large Language Models. ICML 2024.
> > >
> > > [5] Beier Luo et al. Your Pre-trained LLM is Secretly an Unsupervised Confidence Calibrator. NeurIPS 2025.

---

> > > > ### Comment · Reviewer_tumR · 2025-11-26
> > > >
> > > > Thanks for the author's response. I believe the author has addressed most of my concerns. Please remember to add the additional experiments in the final version. However, I have an additional question: if we know the model's confidence before giving the answer, I think this is more like exploring the boundaries of the model's knowledge, rather than a traditional confidence calibration problem. This is because confidence calibration involves calibrating the confidence of a given answer to determine whether to accept the answer provided by the model. I hope the author can share some opinions on this issue, and I will consider improving my score.

---

> ### Author Response · Authors · 2025-11-26
> **Author Response**
>
> Thank you for your quick reply. We are very glad to hear that our response has addressed most of your concerns, and we appreciate your deeper thinking about our work.
>
> As you mentioned, teaching a model to recognize its confidence before generating an answer is indeed a way to enhance its perception of knowledge boundaries. However, we think that if a model can evaluate whether its generated answer is correct, this also reflects a form of knowledge boundary perception.
>
> **Confidence generally falls into two categories**:
>
> 1. **Question-level confidence**: the model’s confidence in answering a question correctly without generating the answer.
>
> 2. **Answer-level confidence**: the model’s confidence in a specific generated answer.
>
> To measure a model’s confidence in a question, the most straightforward method is **semantic uncertainty**[1]: asking the model to generate multiple responses to the same question, comparing their semantic consistency, clustering them, and then using entropy to represent the model’s confidence. However, this method is relatively expensive, as it **requires many rounds of consistency checking**.
>
> It is not difficult to observe that, after clustering, the largest cluster plays the most critical role in entropy computation. The size of this cluster represents the model’s confidence in its most likely answer. We think that **the model’s confidence in its most likely answer essentially reflects its confidence in whether it can answer the question correctly**. Since it is difficult to obtain the model’s most confident answer directly, following prior work[3], we approximate it using the greedy-search answer. This **reduces the complexity of consistency checking from O(n²) to O(n)** where n is the count of responses. We discuss the relationship between self-consistency-based confidence [2-3] and semantic uncertainty [1] in lines 176–183 of the paper.
>
> As you pointed out, this work is fundamentally about helping the model recognize its knowledge boundaries, which is why the title of our paper is **“Honesty Alignment.”** Performing confidence calibration is the way of honesty alignment or enhancing the model’s perception of its knowledge boundaries.
>
> Once again, thank you for reviewing our paper and for your positive reception of our rebuttal. We will incorporate all the content discussed during the rebuttal period into the paper. If you have any further questions, please feel free to contact us.
>
> ---
>
> **References.**
>
> [1] Lorenz Kuhn et.al. Semantic Uncertainty: Linguistic Invariances for Uncertainty Estimation in Natural Language Generation. ICLR 2023
>
> [2] Potsawee Manakul et.al. SelfCheckGPT: Zero-Resource Black-Box Hallucination Detection for Generative Large Language Models. EMNLP 2023
>
> [3] Yuqing Yang et.al. Alignment for Honesty. NeurIPS 2024

---

> > ### Comment · Reviewer_tumR · 2025-11-26
> >
> > Thanks for your response. I have improved my score.

---

> > > ### Author Response · Authors · 2025-11-26
> > > **Author Response**
> > >
> > > Once again, thank you for your positive feedback and for the contributions you have made to improving our paper. This means a great deal to us. Have a nice day!

---

### Author Response · Authors · 2025-12-01
**Thank you for your efforts in the review process!**

Dear AC and Reviewers,

We would like to express our sincere gratitude once again for the effort you have devoted to the review process. The insightful feedback has greatly helped us improve our work. We deeply appreciate the reviewers’ time, careful attention, and detailed comments, as well as their willingness to adjust scores.

Due to the system bug, we fully understand that the AC faces a significant burden, and we are especially grateful for your efforts. To help reduce the AC’s workload, we provide below a concise summary of our paper and the key points from the rebuttal.

---

> **Paper Summary**.

This paper investigates **annotation-efficient universal honesty alignment**, aiming to train language models to **express calibrated confidence that reliably reflects correctness**, thereby improving trustworthiness by making it clear when a model is likely to be right or wrong.

Existing training-based approaches typically rely on **correctness signals** to supervise confidence expression, which **requires ground-truth answers**. As a result, achieving *universally optimal* honesty across tasks demands substantial labeled data.

This paper argues that correctness annotations play two roles: (1) teaching models to express confidence, and (2) calibrating that expressed confidence against correctness. If confidence expression can instead be learned from a reliable unsupervised signal, then only a small amount of correctness supervision is required for calibration.

Building on this insight, the paper proposes **Elicitation-Then-Calibration (EliCal)**, a two-stage training framework:

- **Stage 1: Confidence Elicitation** — Use large-scale self-consistency–based confidence to teach the model to express its internal confidence. Self-consistency provides a reliable training-free signal that aligns well with correctness and is inexpensive to obtain.
- **Stage 2: Confidence Calibration** — Use a small number of correctness labels to calibrate the expressed confidence to actual accuracy.

These stages parallel a **pretraining–finetuning** paradigm.

To support large-scale training and evaluation, the paper introduces **HonestyBench**, a benchmark that unifies ten factual QA datasets (≈560k training pairs and 70k evaluation pairs) annotated with both correctness and consistency signals.

The main findings are:

1. **HonestyBench enables** both EliCal and Calibration-only approaches to reach near **upper-bound alignment** across ten QA tasks when trained on all 560k+ correctness annotations, surpassing the strongest training-free baseline by more than 17%.
2. **EliCal** achieves **≈98% of this upper bound** using only **1k labeled samples** (∼0.18%).
3. **EliCal**, when trained on HonestyBench, consistently delivers substantially better alignment on MMLU than Calibration-only, demonstrating its **superior generalization ability**.

---

> **Rebuttal Summary**.

We are pleased that we were able to address the concerns raised by reviewer `tumR` and reviewer `6wW6`, and we appreciate their positive stance in their final responses. Reviewer `tumR` had already raised the score to **6** before the system issue became widely known, and reviewer `6wW6` noted that *“most concerns are well addressed”* and *“The scores will be adjusted positively when the system allows,”* indicating that the final score would likely be **6**. Due to time constraints, reviewer `wtAu` and reviewer `R9C2` have not yet responded, but we think their concerns could likely be addressed. We are delighted to have ultimately received **all positive scores (6,6,6,6)** and hope that our work will make a meaningful contribution to the community.

---

We are working to incorporate all discussions from the rebuttal period into the appropriate sections of the paper.

Thank you once again to the AC and all the reviewers for your valuable efforts!

Best regards,

Authors

---

### Meta-Review · Area_Chair_sn3K · 2026-01-07

**Summary:**

This work targets honesty alignment for LLMs. They propose a two-stage framework Elicitation-Then-Calibration (EliCal): (1) elicitation of an internal confidence signal using self-consistency from multiple generations without human labels, followed by (2) calibration of the confidence using a small labeled set. The paper also combines HonestyBench, a consolidated benchmark spanning 10 factual QA datasets with correctness labels and consistency signals. EliCal approaches near fully-supervised calibration performance while using a tiny fraction of labels, with some transfer to unseen tasks (e.g., MMLU).

Across reviews, there is agreement that the problem is important and the benchmark + empirical study are valuable. However, there are concerns about technical soundness and evaluation completeness, especially around the central assumption that self-consistency is a robust proxy for confidence, the adequacy of ablations, and missing comparisons/metrics. Authors added lots of experiments results during rebuttal stage, most concerns about their evaluations was addressed.

**Reviewer Concerns:**

1. One reviewer questions whether self-consistency reliably quantifies confidence and asks for more direct evidence and robustness analysis.

2. Ablations and design-factor analysis are incomplete: Key hyperparameters and design choices are underexplored, e.g., impact of the number of sampled generations k, the necessity and marginal gain of Stage 2 (vs elicitation alone), accuracy of the linear head predicting consistency (e.g., MSE / calibration curves), using true self-consistency (computed directly) vs the predicted proxy.

3. Baseline fairness and missing comparisons (major): Reviewers raise concerns that some baselines estimate confidence after producing an answer (Prob/Verb), while EliCal targets pre-generation confidence, potentially creating a mismatch. Multiple reviewers also ask for more recent / stronger calibration baselines (e.g., token-probability / post-hoc calibration methods) and for ECE to be reported broadly, not just for EliCal and Cal-Only.

Authors added lots of new experimental results during rebuttal, solving most of concerns for experiments.

**Reviewer Scores:**

5 (weak accept)

---

### Decision · Program_Chairs · 2026-01-26

Accept (Poster)